# Cortical Parcellation via Spectral Graph Convolutions

**Karthik Gopinath**                                    KARTHIK.GOPINATH.1@ETSMTL.NET
**Christian Desrosiers**                              CHRISTIAN.DESROSIERS@ETSMTL.CA
**Herve Lombaert**                                      HERVE.LOMBAERT@ETSMTL.CA
*ETS Montreal, Canada*

## Abstract

Brain surface analysis is challenging due to the high variability of the cortical geometry. This paper presents a novel graph convolutional based approach for learning surface data directly across multiple surfaces. Current methods either rely on costly geometrical simplification processes or lack the ability to compare surface data across different domains. Our work leverages advances in spectral graph matching to align incompatible surface bases to a reference surface for direct learning of surface data. We illustrate with a cortical parcellation application the benefits of our method. We validate the algorithm over 101 manually labeled brain surfaces. The improvements in parcellation reveal a 29% increase in accuracy with drastic speed gains over conventional methods. The proposed method can be applied to other analysis of surface data, particularly relevant for studying neurological disorders.

**Keywords:** Graph Convolution Networks; Geometric Deep Learning; Cortical Parcellation

## 1. Introduction

Statistical frameworks on brain surfaces are of particular interest to neuroscience, due to their key role in cognition, vision and perception. Conventional approaches rely on costly geometrical simplification processes (Tustison et al., 2014). For instance, the widely used FreeSurfer (Fischl et al., 2004) slowly deforms brain models towards labeled atlases, taking around 3 hours to parcellate brain surfaces. State-of-the-art learning methods (Litjens et al., 2017; Kamnitsas et al., 2017; Dolz et al., 2018) are mostly limited to images, on grid-like structures. Recent geometric deep learning methods (Bronstein et al., 2017; Monti et al., 2017) propose to use convolutional filters on irregular graphs offering a drastic speed advantage. The main concern of (Bronstein et al., 2013; Kovnatsky et al., 2013; Eynard et al., 2015; Parisot et al., 2017) is their inability to compare surface data across different surface domains. One approach is to rely on surface parameterization, for instance, by mapping local graph information onto geodesic patches. Recent approaches are, however, fundamentally defined in Euclidean spaces (Masci et al., 2015; Boscaini et al., 2016; Monti et al., 2017).

The proposed approach leverages recent advances in spectral graph matching to transfer surface data across aligned spectral domains (Lombaert et al., 2015a). This spectral alignment was exploited to learn surface data (Lombaert et al., 2015b), but was limited to pointwise information, ignoring local patterns within surface neighborhoods. Our contributions are multifold. Our novel approach enables a direct learning of surface data across compatible surface bases by exploiting spectral filters over intrinsic representations of sur-

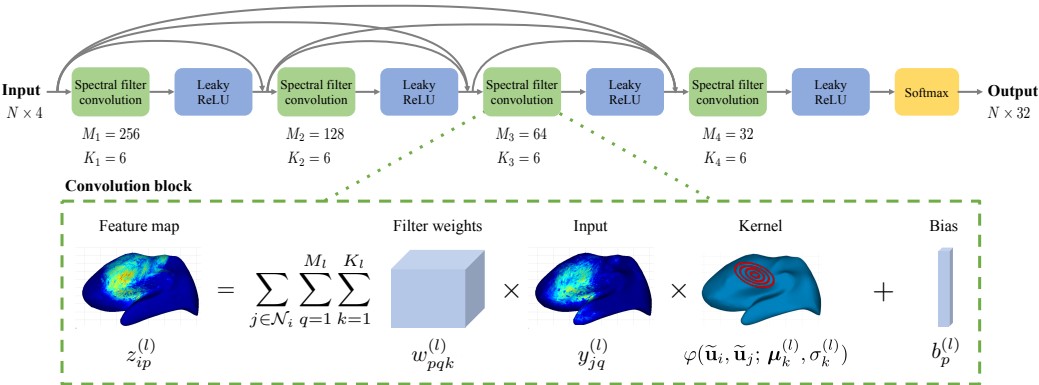

Figure 1: **Overview of the network architecture** – Dense connections are used among successive layers. Weights $(w)$, biases $(b)$, and spectral filters $(\mu, \sigma)$ are learned.

face neighborhoods. Further details are available in (Gopinath et al., 2019). We illustrate the learning capabilities of this approach over 101 manually labeled brain surfaces (Klein et al., 2017) with brain parcellation as an application. Significant improvements of spectral graph convolutions over Euclidean approaches are observed, from Dice scores of 50% to 85%. The accuracy is similar to FreeSurfer (Fischl et al., 2004), scoring 84% (Klein et al., 2017), however the computation gains speed, from hours to seconds.

## 2. Method

An overview of the proposed network is shown in Fig. 1. Firstly, cortical surfaces are modeled as a brain graph $\mathcal{G} = \{\mathcal{V}, \mathcal{E}\}$ , such that $|\mathcal{V}| = N$, and edge set $\mathcal{E}$. Each node $i$ has a feature vector $\mathbf{x}_i \in \mathbb{R}^4$ representing its 3D coordinates and sulcal depth. We map $\mathcal{G}$ to a low-dimension spectral manifold using a normalized graph Laplacian operator $\mathbf{L}$. The eigendecomposition $\mathbf{L}$ is given by $\mathbf{L} = \mathbf{U}\boldsymbol{\Lambda}\mathbf{U}^{-1}$, with the normalized spectral coordinates of nodes as $\widehat{\mathbf{U}} = \boldsymbol{\Lambda}^{-\frac{1}{2}}\mathbf{U}$. The spectral embedding of different brain surfaces are then aligned in the manifold to a reference $\widehat{\mathbf{U}}_{\text{ref}}$ (Gopinath et al., 2019). The optimal transformation between matched nodes is then obtained by iterating until convergence. Finally, a geometric convolutional neural network is used to map input features, corresponding to the spectral coordinates and sulcal depth of brain graph nodes, to a labeled graph. A generalized convolution operation on a graph $\mathcal{G} = \{\mathcal{V}, \mathcal{E}\}$, with $\mathcal{N}_i = \{j \,|\, (i, j) \in \mathcal{E}\}$, as the neighbors of node $i \in \mathcal{V}$, is defined as:

$$z_{ip}^{(l)} \;=\; \sum_{j \in \mathcal{N}_i} \sum_{q=1}^{M_l} \sum_{k=1}^{K_l} w_{pqk}^{(l)} \cdot y_{jq}^{(l)} \cdot \varphi(\widehat{\mathbf{u}}_i, \widehat{\mathbf{u}}_j; \boldsymbol{\Theta}_k^{(l)}) \;+\; b_p^{(l)}, \tag{1}$$

where $\varphi(\widehat{\mathbf{u}}_i, \widehat{\mathbf{u}}_j; \boldsymbol{\Theta}_k)$ is a symmetric kernel in the embedding space with parameter $\boldsymbol{\Theta}_k$. In this work, we follow (Monti et al., 2017) and use a Gaussian kernel: $\varphi(\widehat{\mathbf{u}}_i, \widehat{\mathbf{u}}_j; \boldsymbol{\mu}_k, \sigma_k) = \exp\left(-\sigma_k \,\|(\widehat{\mathbf{u}}_j - \widehat{\mathbf{u}}_i) - \boldsymbol{\mu}_k\|^2\right)$. Using the formulation of Eq. (1), we define a fully-convolutional network with output layer of the network being the number of parcels to be segmented, 32 in our case.

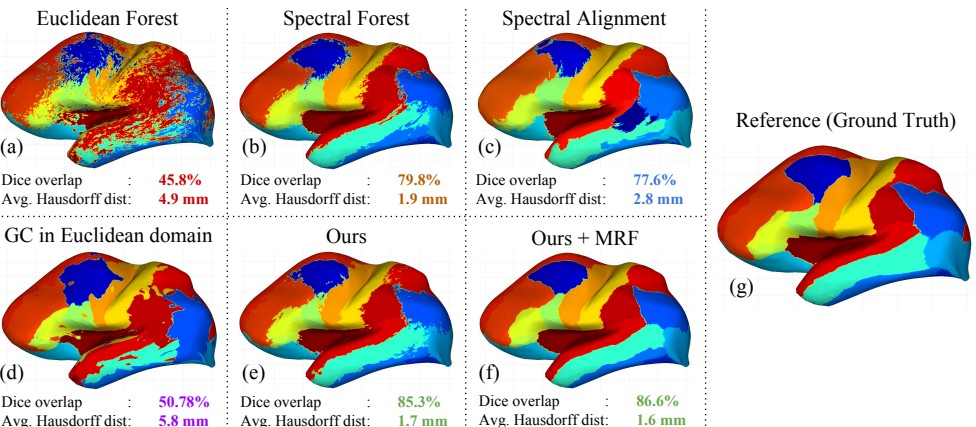

Figure 2: **Cortical Parcellation** – ($a,d$) Learning in Euclidean domian yields low dice score (45.8% with Random Forests, 50.8% with graph convolutions). ($b,e$) Learning with Spectral coordinates: improves Dice score (79.8% with Spectral Forests, 85.3% with our Spectral convolutions). ($c$) A pure spectral alignment without learning yields a Dice score of 77.6%.($f$) MRF regularization leads to an improvement in Dice score (86.6%) and boundary regularity. ($g$) Reference ground truth for comparison purposes. Brain surfaces are inflated for visualization.

## 3. Results

First, we assess the improvement in accuracy of learning frameworks when operating directly in a spectral domain rather than a conventional Euclidean domain. The effect of learning over a spectral domain is also assessed using, first, pointwise information in the Random Forest framework and in graph convolutional networks. Fig. 2 shows that indeed learning using spectral method produces an improved parcellation quantitatively.

Second, we highlight the advantages of spectral alignment in this framework. We train and test our algorithm with 5 different reference brains to verify the independence of our method with respect to the choice of a reference for alignment. The evaluation shows a similar performance for all references with an average dice score of 86.4% and a standard deviation of only 0.17%, indicating robustness to the choice of references.

Shifting graph convolutions into a spectral domain endows the learning process with a geometry-aware representation of surface data with classification improving from a 50% Dice score in a conventional Euclidean domain to an 85% Dice score in a spectral domain. An extra improvement of 29% is also gained by exploiting spectral neighborhoods from 50% to 79%.

## 4. Conclusion

This paper presents a novel framework for learning surface data via spectral graph convolutions. This is a particularly challenging problem where current graph convolution approaches remain limited by the inability to compare surface data across brain geometries. The algorithm leverages recent advances in spectral matching to enable such comparisons. While the potential of our method was demonstrated on cortical parcellation, it can be applied to other analyses of surface data, potentially leading to new families of geometry-based biomarkers for neurological disorders.

## Acknowledgements

This work was supported financially by the Natural Sciences and Engineering Research Council of Canada (NSERC). We also gratefully acknowledge the support of NVIDIA Corporation with the donation of a Titan Xp GPU used for this research.

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
