# OpenReview forum: "Cortical Parcellation via Spectral Graph Convolutions"
_MIDL.io/2019/Conference/Abstract — MIDL Abstract 2019_

### Official Review · AnonReviewer2 · 2019-04-24
**Overall a very good abstract, fitting perfectly the MIDL abstract guidelines.**

**Rating:** 4
**Confidence:** 2

**Review:**

This abstract presents a novel framework for learning surface data via spectral graph convolutions. The work is well written and clearly structured.

This work was apparently under review for MICCAI 2018 and has been recently published in medical Image Analysis. The journal version is clearly cited.

Further potential of this approach remains vague, just outlining that work like this can "potentially leading to new families of geometry-based
biomarkers for neurological disorders".

Overall a very good abstract, fitting perfectly the MIDL abstract guidelines.

---

### Official Review · AnonReviewer1 · 2019-04-25
**good summary of recently published work**

**Rating:** 3
**Confidence:** 2

**Review:**

This well-written abstract summarizes a recent MedIA articles recently published by the same authors.
The motivation of the work is cortical parcellation. The authors propose mapping the brain surface mesh coordinates and a corresponding set of scalar measurements (e.g., sulcal depth) to a low-dimensional spectral manifold, and then using a geometric CNN to learn labels from spectral coordinates and the scalar measurements, with compelling results.
I believe that this poster could lead to good discussion at the conference.

---

### Decision · Program_Chairs · 2019-05-06
**Acceptance Decision**

Accept